# Use of Continuous Positive Airway Pressure (CPAP) to Limit Diaphragm Motion—A Novel Approach for Definitive Radiation Therapy for Inoperable Pleural Mesothelioma: A Pilot Study

**DOI:** 10.3390/biology10080711

**Published:** 2021-07-24

**Authors:** Assaf Moore, Marc J. Kindler, Aaron Max Allen

**Affiliations:** 1Department of Radiotherapy, Institute of Oncology, Davidoff Cancer Center, Rabin Medical Center—Beilinson Hospital, Petach Tikva 4941492, Israel; moorea4@mskcc.org (A.M.); yonik@clalit.org.il (M.J.K.); 2Sackler Faculty of Medicine, Tel Aviv University, Tel Aviv 6997801, Israel

**Keywords:** mesothelioma, VMAT, radiotherapy

## Abstract

**Simple Summary:**

Radiotherapy is an important part of the multimodality approach to treating malignant pleural mesothelioma. In recent studies there is a new trend to treat patients with intact lungs instead of following surgery. This treatment creates significant concerns regarding lung toxicity. We describe two methods to reduce that toxicity. One is the use of constant pulmonary airway pressure (CPAP) to inflate the lungs during treatment. The second is utilizing a novel method of planning and delivering radiotherapy called volumetric modulated arc therapy (VMAT).

**Abstract:**

Malignant pleural mesothelioma (MPM) is a deadly disease and radiotherapy (RT) plays an important role in its management. Recent developments in technique have made it is possible to deliver RT to MPM in the intact lung. However, it is imperative to reduce normal lung doses. We present a pilot study examining the use of CPAP and VMAT radiotherapy to reduce toxicity when treating MPM, involving three consecutive patients with MPM, not amenable to surgery, who were treated according to Helsinki committee approval. Patients were simulated using four-dimentional CT simulation with the assistance of CPAP lung inflation, then were treated using both IMRT and VMAT techniques. Radiation lung dose was optimized based on accepted lung dose constraints. Patients were followed for toxicity as well as local control and survival. Results: Three patients were treated with CPAP-based IMRT treatment. These patients tolerated the treatment and DVH constraints were able to be met. The comparison plans among the four VMAT arcs and the IMRT static field treatment were able to accomplish the treatment planning objectives without significant advantages with either technique. The treatment combined with CPAP reduced the normal lung dose in MPM patients with intact lungs. This technique is worthy of further investigation.

## 1. Introduction

Malignant pleural mesothelioma (MPM) is a rare and aggressive thoracic malignancy with an established association with occupational or environmental asbestos exposure. About 3000 new cases are diagnosed each year in the US [1]. The management of these patients should involve an experienced multidisciplinary team [2,3]. In general, the majority of MPM patients require more than one treatment modality. These modalities include surgery, radiation therapy (RT), and systemic chemotherapy.

RT can be used as a part of multimodality treatment, as a single modality with definitive or palliative intent. In the postoperative setting, hemithoracic radiation following EPP seems feasible and may decrease local recurrence rates [4,5,6,7,8,9,10]. A neoadjuvant approached has also been explored [11,12,13,14].

Breathing-related tumor and organ motion constitute significant problems when planning RT for tumors of the chest and upper abdomen, and accounting for it complicates treatment planning and delivery. A major concern with hemithoracic RT is crippling or fatal pneumonitis. [10], The relative radiosensitivity of the normal thoracic organs limits the delivery of an effective RT dose to the pleural surfaces. In addition, treating the entire involved pleura renders sparing of the ipsilateral lung challenging.

Several means of decreasing pulmonary toxicities have been explored in MPM, including intensity-modulated radiation therapy (IMRT) based planning [6,15,16], intensity-modulated proton therapy (IMPT) [17], and restricting the IMRT field [18].

Continuous positive airway pressure (CPAP) devices have historically been used in patients with obstructive sleep apnea to maintain airway patency. Some of the physiological effects noted during CPAP are hyperinflation of the lungs, stabilizing and flattening of the diaphragm, and decreasing the tidal volume. It has been previously shown that CPAP can increase lung volumes and decrease the mean lung dose (MLD) and V5 [19]. CPAP may also decrease tumor motion and, thus, the internal target volume (ITV) [20].

In light of the above, the use of CPAP to control respiratory motion seems attractive for the treatment of MPM.

We initiated a prospective study to evaluate CPAP performance in reducing tumor motion in lung and upper abdomen cancers, improve treatment geometry, and reduce heart exposure for patients with left breast cancers [21].

Volumetric modulated arc therapy (VMAT) is an advanced type of IMRT that has several advantages, including a more rapid treatment delivery, fewer monitor units (MU) used, and improved optimization with highly conformal dose distribution with target volume coverage.

In this report, we summarize our experience with utilizing CPAP in the definitive treatment of unresectable or inoperable MPM patients in combination with modern radiotherapy techniques. We also attempted an additional sub-study to re-plan patients who were treated with standard IMRT-based RT with VMAT, in order to assess its potential benefits in these complex cases.

## 2. Materials and Methods

### 2.1. Patients

This report includes three consecutive patients with biopsy-proven MPM who were recruited to the CPAP study at the Davidoff Cancer Center (DCC) at the Rabin Medical Center (RMC), and treated with definitive RT and CPAP breathing control.

### 2.2. Data Collection and Outcomes

This study was approved by the medical center’s institutional Helsinki review board. All patients signed a written informed consent form, according to ICH-GCP. Data were collected from medical records and included demographics, medical comorbidities, location and extent of disease, imaging findings, radiation treatment details, follow-up and other procedures in the Pulmonology Institute, performance status, response to treatment, survival, and cause of death. Staging was based on the 8th edition of the American Joint Committee on Cancer (AJCC) staging manual.

### 2.3. CPAP Simulation

Patients enrolled in this trial were trained to facilitate the use of daily CPAP-assisted radiotherapy. Prior to simulation, patients wore the CPAP mask (Weinman Prisma) to acclimatize them to positive pressure. Every patient underwent pulmonary function tests and respiratory clearance for CPAP prior to initiation of CPAP. Initial CPAP pressure was chosen at 4 mmHg, and gradually elevated according to patient comfort to reach a goal pressure of 15 mmHg.

### 2.4. Treatment Planning

EBRT Patients were immobilized for simulation with a customized vacuum cushion for CT simulation. The gross tumor volume (GTV) was defined as the extent of macroscopic disease by imaging studies, which for the purposes of this study was GTV = clinical target volume (CTV). The planning treatment volume (PTV) was defined as a 5-mm margin around the GTV. The PTV was reduced in cases of proximity to vital normal tissue. Patients were treated with IMRT using dynamic sliding window multileaf collimator (MLC).

For study purposes only, patients were re-planned with VMAT in a 4-arc configuration. Specification of the dose-volume histogram (DVH) constraints is available in Table 1. Dose calculations were performed using the Eclipse™ treatment planning system, AAA algorithm version 8 (Varian Medical Systems, Inc., Palo Alto, CA, USA). Treatment was originally prescribed to the 95% isodose line with a PTV tolerance of ±5%, and modified by tissue tolerance. Quality assurance verification plans were performed with the ArcCHECK™ dosimeter (Sun Nuclear Corporation, Melbourne, FL, USA).

### 2.5. Post RT Evaluation

Treatment outcomes were assessed clinically by symptoms assessment and pulmonary function tests, as well as by follow-up bronchoscopies every 3–6 months. Computed tomography (CT) scans were performed when appropriate.

## 3. Results

### 3.1. Patients

Three consecutive cases were included. Before being referred for definitive radiotherapy, all cases were discussed in a multidisciplinary team that included thoracic surgeons, pulmonologists, medical oncologists, radiation oncologists and radiologists. RT treatment parameters are summarized in Table 2.

#### 3.1.1. Patient 1

A 75-year-old woman, past smoker (10 pack years), with a known history of hypertension and hyperlipidemia, no asbestos exposure, who was diagnosed in June 2017 with biphasic MPM in the right hemithorax. A fluorodeoxyglucose (FDG)-positron emission tomography–computed tomography (PET/CT) showed high FDG uptake in the parietal, visceral, and mediastinal pleura, as well as focal involvement of the right chest wall, T3N0M0, stage IB. The patient was determined to be inoperable by a multidisciplinary team and was referred for definitive radiotherapy. The patient was treated to 60 Gray (Gy) between August and September 2017. During RT, the patient developed grade 2 nausea and vomiting, grade 3 thrombocytopenia due to suspected immune thrombocytopenia (ITP), and improved after completion of RT and maintenance prednisone. Following RT, the patient developed persistent grade 3 pneumonitis. Disease remained stable until a PET/CT in March 2018 demonstrated high FDG uptake in suspected liver metastases. A biopsy was positive for metastatic MPM. The patient received systemic treatment and died in November 2018. The pleural disease remained stable until August 2018.

#### 3.1.2. Patient 2

A 62-year-old man, past smoker (35 pack years), known history of diabetes type II and hyperlipidemia, who worked in asbestos production, was diagnosed in May 2017 with epitheloid type MPM in the left hemithorax. A PET/CT showed high FDG uptake in the parietal, viscera, and mediastinal pleura, as well as in mediastinal and internal mammary nodes, T2N2M0, stage IIIB. The patient refused surgery and was treated with definitive RT to dose of 54 Gy between September and October 2017. The patient did not experience any adverse events. A PET/CT in December 2017 demonstrated a partial response. The patient progressed clinically, as well as in a PET/CT, in August 2018 and began treatment with carboplatin-pemetrexed in November 2018. At the time of preparation of this manuscript, the patient is alive and continues palliative systemic treatment.

#### 3.1.3. Patient 3

A 65-year-old man, non-smoker, known history of type II diabetes, hypertension, hyperlipidemia and ischemic heart disease, no history of asbestos exposure, diagnosed in May 2017 with epitheloid type MPM of the right hemithorax. A PET/CT showed high FDG uptake in pleural masses, as well as in mediastinal, supra and infraclavicular nodes T2N2M0, stage IIIB. The patient enrolled in a clinical trial investigating the addition of a tyrosine kinase inhibitor to standard chemotherapy in May 2017. Disease progressed in December 2017. The patient was then referred for RT and treated to 54 Gy between January and February 2018 and remained without progression until July 2018—with the appearance of distant nodal involvement and peritoneal spread. The thoracic disease remained stable until late September 2018—when one of the pleural masses increased in size. At the time of preparation of this manuscript, the patient is alive and continues palliative systemic treatment.

A comparison of dosimetry between IMRT step and shoot and 4 ARC VMAT is presented in Table 3. In addition Figure 1 shows a representation of the planning.

One can see a comparison of the effects of CPAP on the planning of the radiotherapy in these patients as is shown in Figure 2.

These anatomic changes provided with CPAP inflation lead to improved dose distributions as seen in Figure 3.

## 4. Discussion

In this study, we described our experience with three MPM patients treated with definitive radiotherapy and CPAP breathing control. To put this experience in context it is critical to examine the changing role of RT in MPM.

While much data has been reported regarding RT following EPP little data is available for the non-operable patient. Evidence supports the use of palliative RT in non-operable MPM. A dose of 20 Gy in five fractions achieved pain relief in almost 50% of patients in a phase II trial [22]. A dose of 4 Gy per fraction may be associated with higher local response rate compared with lower doses [1]. In a report of palliative RT to a dose of 36 Gy for the treatment of 54 MPM patients, the radiologic response rate was 43%. There was one case of grade 3 radiation pneumonitis, and one case of grade 3 nausea and vomiting. The median survival from diagnosis was 11.3 months [23].

There are no randomized control trials assessing the benefit of definitive RT in nonoperable MPM, however, retrospective data exist. In a retrospective study using data from the National Cancer Data Base (NCDB), definitive RT of 40–65 Gy was associated with improved survival (hazard ratio (HR) = 0.77) on multivariable analysis in patients with invasive, non-metastatic pleural mesothelioma [24]. Compared to patients who did not undergo surgery or receive definitive RT, surgery and RT combined was associated with a greater decrease in mortality (HR = 0.60) than surgery alone (HR = 0.75) or RT alone (HR = 0.74) [24]. The optimal dose for MPM has not been determined. [25]

All patients in our cohort were treated with IMRT based planning. IMRT has been employed in MPM in the post-operative setting as means of decreasing V20, V5, and the mean lung dose [16,26]. In one report, patients with fatal pneumonitis had a V20 of 15.3%–22.4%, V5 of 81–100%, and mean lung dose of 13.3–17 Gy [27]. In another report, V20 was the only factor that predicted pulmonary related deaths with IMRT. A V20 > 7% had a 42-fold risk [16]. In initial experiences, intensity modulated proton therapy (IMPT) has shown feasibility and lower mean doses to the contralateral lung, heart, esophagus, liver, and ipsilateral kidney compared with IMRT [17,28,29,30]. Dose escalation has not been shown to significantly reduce local failure after EPP [15,30,31,32].

We demonstrated that CPAP inflated the thorax, displaced the heart and liver away from RT fields, and increased lung volume as was shown in Figure 2.

Additionally, a more inferiorly displaced liver with CPAP in comparison to FB can reduce unnecessary radiation exposure to the liver in right-sided RT plans as was shown in Figure 3.

CPAP as compared to FB can minimize breathing-caused thoracic movements, thus resulting in the radiation target (whole breast or chest wall) becoming more stable and suitable not only for 3DCRT, but also for high-precision RT (IMRT, VMAT). An important future research direction will be to compare clinical outcomes between IMRT and VMAT treatment such as monitoring for differences in pulmonary function testing following RT with differing techniques.

The use of CPAP is novel for this indication, although some evidence exists in other clinical scenarios. In a prospective trial of patients with primary or secondary lung tumors referred for SBRT and simulated with free breathing and CPAP, the use of CPAP significantly decreased tumor excursion in all plains [20]. In another study of CPAP for lung SBRT, the use of CPAP was found to significantly increase lung volume and decrease MLD and V5 with no impact on MHD. The authors concluded that the slight decrease in radiation dose to the lungs would likely be clinically insignificant [19]. CPAP has also been studied in breast cancer [33]. In a study of three patients who were unable to maintain and reproduce DIBH, stimulated with FB vs. CPAP, CPAP significantly increased the total lung volume and increased distance from the sternal notch to the superior portion of heart by 0.5–1.25, the heart volume within the left-sided tangential fields was substantially decreased by more than 92%. The inflated thorax with CPAP also displaced the liver, at least 2 cm inferiorly [34].

## 5. Conclusions

RT is effective for the management of pleural mesothelioma and can offer long-term local control. The use of CPAP may improve dosimetry and reduce toxicity and should be further explored.

## Figures and Tables

**Figure 1 biology-10-00711-f001:**
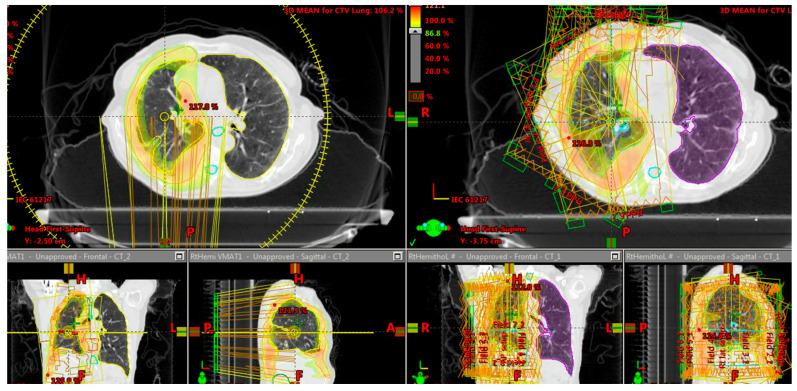
Demonstrates the difference in dose distribution graphically between the two plans. On the left we show axial, coronal and sagittal views of VMAT planning. On the right side the same views with IMRT step and shot planning. In both views the ability to spare the internal lung tissue is seen with slight advantage to VMAT planning.

**Figure 2 biology-10-00711-f002:**
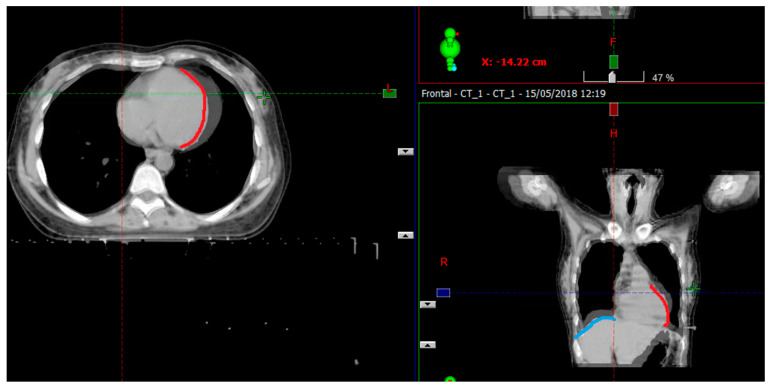
Shows overlay of CPAP and non-CPAP scans. The blue line the difference in diaphragm position and the red line the difference in heart position.

**Figure 3 biology-10-00711-f003:**
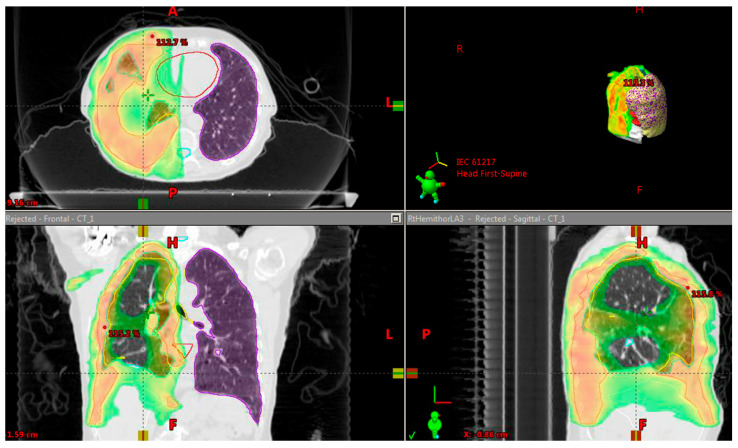
Showing the RT dose distribution with CPAP lung inflation allowing sparing of lung and cardiac tissue.

**Table 1 biology-10-00711-t001:** DVH constraints for organs at risk.

Organ	Constraints
Lung	Mean dose < 18 Gy V5 < 60% V20 < 35%
Spinal Cord	Max dose < 50 Gy
Esophagus	Mean dose < 55 Gy
Heart	Mean dose < 15 Gy

Abbreviations: Gy, Gray; V5, proportion of the lung receiving 5 Gy; V20, proportion of the lung receiving 20 Gy.

**Table 2 biology-10-00711-t002:** Patient and tumor characteristics, treatment parameters, and outcomes.

Items	Patient 1	Patient 2	Patient 3
Histology	biphasic	epitheloid	epitheloid
Stage	T3N0M0, stage IB	T2N2M0, stage IIIB	T2N2M0, stage IIIB
PS at RT	1	1	1
Dose	54 Gy	54 Gy	54 Gy
coverage	95% dose to 95% target volume	90% dose to 90% target volume	95% dose to 90% target volume
Clinical target volume (cc)	1852.31	1718.23	1500.00
Ipsilateral lung volume (cc)	1218.07	691.66	1418.07
Ipsilateral lung V20 (%)	99.7%	85.9%	85.9%
Ipsilateral lung V5 (%)	100%	100%	100%
Contralateral lung volume (cc)	2511.95	1692.7	2059.36
Contralateral lung V20 (%)	0.5%	0%	0%
Contralateral lung V5 (%)	46.9%	20.8%	36.4%
Both lungs volume (CC)	3957.04	2375.76	3485.35
MLD	17.051	11.3	17.06
Both lungs V20 (%)	33%	25%	35.3%
Both lungs V5 (%)	64.7%	44.1%	62.57%
Mean Heart Dose (Gy)	24.29	15.85	13.98
Heart V30	40.8%	2.3%	9.9%
Mean Liver Dose (Gy)	23.36	5.39	20.4
Duration of response	Started—8/1/17 Completed—9/10/17 Progressed—5/23/18 (distant), 8/12/18 (local)	Started—9/25/17 Completed—10/31/17 Progressed—8/26/18	Started—1/16/18 Completed—2/21/18 Progressed—9/26/18

**Table 3 biology-10-00711-t003:** Dosimetric comparison between IMRT and VMAT-based planning.

Items	IMRT	VMAT 4 arcs
**Patient 1**		
Max dose	121.1%	127.3%
Max dose (in CTV)	121.1%	127.3%
Min dose (in CTV)	7.6%	6.4%
Ipsi lung V5	100%	100%
Ipsi lung V20	99.8%	96.5%
Contra lung V5	47.5%	76.6%
Contra lung V20	0.48%	2.6%
MLD	18.9	17.9
Both lungs V5	65.9%	85.1%
Both lungs V20	36.4%	36.7%
Mean Heart Dose	24.3	25.5
Heart V30	40.7%	40.1%
Mean Liver Dose	23.4	28.1
**Patient 2**		
Max dose	153.8%	115.5%
Max dose (in CTV)	147.5%	115.5%
Min dose (in CTV)	34.8%	60.7%
Ipsi lung V5	100%	100%
Ipsi lung V20	94.8%	87.2%
Contra lung V5	26.7%	38.6%
Contra lung V20	0.2%	0.1%
MLD	12.5	14.7
Both lungs V5	47.9%	57.1%
Both lungs V20	27.3%	25.4%
Mean Heart Dose	17.5	20.4
Heart V30	5.6%	10.9%
Mean Liver Dose	5.9	5.5
**Patient 3**		
Max dose	120.4%	119.4%
Max dose (in CTV)	120.4%	115.9%
Min dose (in CTV)	33.8%	71.3%
Ipsi lung V5	100%	100%
Ipsi lung V20	85.5%	93.7%
Contra lung V5	36.4%	45.1%
Contra lung V20	0%	0.2%
MLD	19.6	21.4
Both lungs V5	65.26%	69.5%
Both lungs V20	39.7%	43.0%
Mean Heart Dose	13.9	16.5
Heart V30	9.9%	7.8%
Mean Liver Dose	32.9	19.9

## Data Availability

All data can be found in the electronic medical records of Rabin Medical Center.

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
