# Peer review of "Use of Continuous Positive Airway Pressure (CPAP) to Limit Diaphragm Motion—A Novel Approach for Definitive Radiation Therapy for Inoperable Pleural Mesothelioma: A Pilot Study"

_biology, 2021, doi:10.3390/biology10080711_

Round 1
Reviewer 1 Report
In this manuscript the authors claimed they evaluated 2 methods used for lung toxicity reduction in definitive radiation therapy for inoperable pleural mesothelioma. Two methods are 1) use of CPAP for breathing motion management, and 2) VMAT technique for beam delivery.
Although the subject is not obsolete and may get attention from readers the study was structured and/or described poorly. For example, while the title implies quantitative analysis on motion reduction + dosimetric gain with CPAP over non-CPAP, no such assessment was provided. In addition, the number of patients were too small to obtain meaningful statistical power.
Reviewer 2 Report
Overall Evaluation:Thanks for giving me to review this manuscript.
This is a case series on the innovations in radiotherapy for mesothelioma patients. Authors used CPAP to keep lungs inflated and to reduce PTV.
This is a well-written manuscript. However, there are several methodological aspects that warrant further clarification may be indicated.
Major-Authors showed theoretical superiority of VMAT against conventional IMRT in Table2. They showed imaging superiority in Figure 2. However, they did not show information on long-term prognosis well. Can they show respiratory function and CT images of lung fibrosis after radiotherapy?
-What type of CPAP mask did authors use?
-Authors should explain each figure in figure 1-3. For example, figure 1 contains 6 figures without explanations.
-p6 line 170-184 These two paragraphs are background knowledge, not discussion about the results of this manuscript.
-Figure 2 and 3 should be moved to the results or methods.
Round 2
Reviewer 1 Report
I am still not convinced with the response. I assume you must have non-CPAP CT data set for 3 patients. If it is the case, you can make plans with non-CPAP CT data sets (probably both IMRT and VMAT) and then compare them with plans with CPAP. The amount of reduction should be reported as well.
Reviewer 2 Report
Authors improved their manuscript. But I want them to revise again.
Authors showed theoretical superiority of VMAT against conventional IMRT in Table2. They showed imaging superiority in Figure 2. However, they did not show information on long-term prognosis well. Can they show respiratory function and CT images of lung fibrosis after radiotherapy?
We highly appreciate the reviewers time and effort in assisting us to make the manuscript better. Regarding changes in respiratory function and lung fibrosis imaging – this study compared the planning of VMAT to IMRT in the planning system only. We do not therefore have comparative imaging/ pulmonary function tests to compare between these approaches. We agree that with a larger sample size and or head to head clinical comparison these data would be helpful however, they are not currently available.
Authors should emphasis this point as a research implication.
